# Distributed dynamic event-triggered algorithm for optimization problem with time delay

**Lu Jiang, Lunchao Xia\*, Zhongyuan Zhao** *\**

College of Automation, Nanjing University of Information Science and Technology, Nanjing, China

\* xialunchao@nuist.edu.cn (LX); zhaozhongyuan@nuist.edu.cn (ZZ)

## Abstract

This paper focuses on studying the optimization problem of multi-agent systems (MAS) under undirected graph. To reduce the communication frequency among agents, a zero-gradient-sum (ZGS) algorithm based on dynamic event-triggered (DET) mechanism is investigated. The event-triggered condition of each agent only uses its own state information and the neighbor's state information at the previous triggering instants, without requiring continuous state information from the neighbor. In addition, the designed algorithm allows for the sampling period to be arbitrarily large. The Lyapunov method is utilized to derive the sufficient conditions that incorporate time delay and parameters. As the event is only checked at the periodic moment, zeno behavior can be directly excluded. Finally, numerical simulations demonstrate the effectiveness of the theoretical results.

**Data Availability Statement:** All relevant data are all within the manuscript.

**Funding:** This research was supported by the National Natural Science Foundation of China (U23B2061), the Natural Science Foundation of

## 1 Introduction

In this paper, a MAS that contains $n$ agents is considered. The objective is to solve the optimization problems given below, where each agent $i$, $i = 1, \ldots, n$, has a cost function $g_i$.

$$x^* = \arg \min_{x \in \Omega} G(x). \tag{1}$$

where $G(x) = \sum_{i=1}^{n} g_i(x)$ and $x^*$ denotes the global optimal value. Many distributed optimization (DO) tasks in industry and economy can be transformed into Problem (1), such as sensor scheduling [1, 2], robot fleeting [3–5], smart grid [6, 7], etc. Thus problem (1) gained extensive research.

In recent years, numerous DO algorithms have been proposed, with the majority of them being based on a discrete iterative form. However, discrete-time optimization algorithms are no longer sufficient for solving problems such as signal processing, parameter estimation, and filter design. In such cases, continuous-time optimization algorithms provide the necessary means. Furthermore, the existing convex optimization theory and Lyapunov stability theory offer powerful tools for the design and convergence analysis of continuous-time algorithms. A DO algorithm based on adaptive mechanism was proposed in [8] to address the issue of unknown network connectivity. By introducing a dynamic coupling gain, the algorithm

Jiangsu Province, China (BK20200824), and the Postgraduate Research and Practice Innovation Program of Jiangsu Province, China (SJCX230391)..Lunchao Xia(SJCX230391): study design, data collection and analysis, and preparation of the manuscript. Zhongyuan Zhao (U23B2061, BK20200824): study design, data collection and analysis, decision to publish, and preparation of the manuscript.

**Competing interests:** The authors have declared that no competing interests exist.

achieves global asymptotic convergence. A scaled integration protocol that employs projection output feedback was presented in [9] to address distributed convex optimization problems with general constraints. A Newton-Raphson consistency algorithm was proposed in [10] to solve the unconstrained DO problem. Under appropriate assumptions, the algorithm can achieve exponential convergence by using second-order information. Among the numerous optimization algorithms, the ZGS algorithm has garnered significant attention from scholars due to its efficient performance in handling optimization problems. A continuous-time ZGS was proposed in [11] to solve the optimisation problem under undirected graphs, and a lower bound on the rate of convergence associated with the network structure was given. A distributed ZGS algorithm was proposed in [12] to solve optimization problems in time-varying networks. This is the further extension of the work in [11].

However, the aforementioned work requires agents to communicate continuously with their neighbors, which will cause a large burden in in practical applications. The ET controllers were first proposed in [13] to address the MAS consistency problem. The ET mechanism involves designing an error-based trigger condition that requires communication between agents when the condition is satisfied. When the condition is not satisfied, communication between agents is not required. An ET subgradient optimization algorithm based on communication network edges was presented in [14], and the exponential convergence rate of the algorithm was given. A ZGS algorithm that utilizes the ET mechanism was introduced in [15] to ensure the system state converges to the optimal solution. However, the aforementioned ET policy is a static ET policy, meaning that its trigger threshold is only related to the state of the agent. As the agents' states gradually converge, the trigger conditions become easily satisfied, leading to unnecessary communication. Therefore, a more reasonable trigger condition needs to be designed. A recursive convex optimization algorithm based on DET mechanism was proposed in [16] to schedule the information transfer between sensor networks. A DET controller was proposed in [17] to adjust the data transmission between sensors. The use of the DET mechanism allows the trigger condition to change with the agent state, thus further reducing the frequency of communication.

As time delay is inevitable in real systems, there has been extensive research on time-delayed ET strategies [18–21]. Motivated by the work in [15], we propose a distributed dynamic event-triggered zero gradient sum (DET-ZGS) algorithm. Compared to most papers that use the ET mechanism [22–25], adopting the DET mechanism can avoid continuous triggering caused when the agent state approaches the optimal value. Considering that it takes some time to conduct the DET judgment, using the previous value to construct the DET condition is more realistic. Moreover, the developed algorithm accommodates an arbitrarily large sampling period. Furthermore, we consider communication time delay and derive explicit necessary conditions by using the Lyapunov method. Finally, Numerical simulations further illustrate the effectiveness of the proposed algorithm.

The overall structure of this paper is arranged as follows. The basic graph theory and knowledge of convex functions are provided in Section 2. The DET-ZGS condition is introduced in Section 3, and the necessary conditions for convergence are given. The simulation results for the two examples are provided in Section 4. Section 5 offers a summary of this paper.

## 2 Preliminaries and problem formulation

### 2.1 Graph theory

$\mathbb{R}$ and $\mathbb{R}^d$ represent the set of real numbers and a vector set respectively. A graph $\mathcal{G} := (\mathcal{V}, \mathcal{E})$ is employed to model a communication network with $n$ nodes. The graph consists of a vertex set $\mathcal{V} = \{1, 2, ..., N\}$ and an edge set $\mathcal{E} \subseteq \mathcal{V} \times \mathcal{V}$. If there exists a path between any two nodes,

then $\mathcal{G}$ is connected. Define $\mathcal{A} = [a_{ij}]$ as the weighted adjacency matrix of $\mathcal{G}$, $a_{ij} > 0 \Leftrightarrow (i, j) \in \mathcal{E}; a_{ij} = 0 \Leftrightarrow (i, j) \notin \mathcal{E}$. $\mathcal{D}$ denotes the degree matrix, where $\mathcal{D} = diag\{d_1, ..., d_n\}$. The Laplace matrix is equal to the degree matrix minus the adjacency matrix, i.e. $\mathcal{L} = \mathcal{D} - \mathcal{A}$. When the graph $\mathcal{G}$ is an undirected graph, its Laplacian matrix is a symmetric matrix.

## 2.2 Convex function

If there exists some positive constant which makes the following inequality hold, then the cost function $g_i : \mathbb{R}^n \to \mathbb{R}$ can be called strongly convex [11].

$$g_i(b) - g_i(a) - \nabla g_i(a)^T (b - a) \geq \frac{\phi_i}{2} \|b - a\|^2, \forall a, b \in \Omega, \tag{2}$$

$$(\nabla g_i(b) - \nabla g_i(a))^T (b - a) \geq \phi_i \|b - a\|^2, \forall a, b \in \Omega, \tag{3}$$

$$\nabla^2 g_i(a) \geq \phi_i I_n, \forall a \in \Omega, \tag{4}$$

where $\phi_i$ is a positive constant, $\nabla g_i$ and $\nabla^2 g_i$ are the gradient and the Hessian matrix respectively.

**Lemma 1** *Suppose the graph $\mathcal{G}$ is strongly connected and balanced, for any $\mathcal{X} \in \mathbb{R}^n$, the following property holds.*

$$\mathcal{X}^T \mathcal{L} \mathcal{X} \geq \frac{\alpha_2}{\beta_n} \mathcal{X}^T \mathcal{L}^T \mathcal{L} \mathcal{X}, \tag{5}$$

*where $\alpha_2$ is the smallest positive eigenvalue on $\frac{\mathcal{L} + \mathcal{L}^T}{2}$ and $\beta_n$ is the largest eigenvalue on $\mathcal{L}^T \mathcal{L}$.*

# 3 The distributed DET-ZGS algorithm and main results

## 3.1 The distributed DET-ZGS algorithm with time delay

In this paper, we adopt a periodic sampling method to obtain the state of the agent, where the sampling period is defined as $h$. This means that we sample the state of the agent at intervals of $h$. Specifically, we use $x_i(t)$ to represent the state of the agent at time $t$, and this state changes dynamically over time. Firstly, the states of the neighbors and the agent's own state at the previous triggering time are sent to the controller to obtain the agent's current state. Simultaneously, an event detector is installed on each agent, which only checks if the triggering condition is met at the sampling time. Subsequently, the agent's state from the previous sampling time is sent to a constructed trigger for evaluation. When the preset triggering condition is satisfied, the agent's state at the triggering time is obtained. Finally, the obtained local state is used to update the control actions of both the agent itself and its neighbors. In practice, when the value at the sampling point exactly meets the DET condition, we have $x_i(t) = x_i(\varsigma h) = x_i(t_k^i)$. Due to the existence of time delay in practical transmission, this section considers satisfying a delay of $\tau \in (0, h)$. In addition, for $t \in [\varsigma h + \tau, (\varsigma + 1)h + \tau), \varsigma = 0, 1, 2, \ldots$, we define $\hat{x}_i(t) = x_i(t_k^i)$ and deviation variable $\delta_i(t) = x_i(t) - \hat{x}_i(t)$. The main structure

of the algorithm is designed as follows:

$$\dot{x}_i(t) = (\nabla^2 g_i(x_i(t)))^{-1} u_i(t), \tag{6}$$

$$u_i(t) = \sum_{j=1}^{n} a_{ij}(\hat{x}_j(t-\tau) - \hat{x}_i(t-\tau)). \tag{7}$$

Then the DET condition is designed as follows:

$$\omega_i(d_i \delta_i^2(\zeta h) - \gamma q_i(\zeta h - h)) \leq \xi_i(\zeta h), \tag{8}$$

where

$$q_i(t) = \sum_{j=1}^{n} a_{ij}(\hat{x}_j(t-\tau) - \hat{x}_i(t-\tau))^2, \tag{9}$$

and

$$\dot{\xi}_i(t) = -\mu_i \xi_i(\zeta h) + \varepsilon_i(\gamma q_i(\zeta h - h) - d_i \delta_i^2(\zeta h)), \tag{10}$$

where $d_i$ is the degree of the agent $i$, $\gamma$ is a positive constant. $\varepsilon_i$, $\mu_i$, $\omega_i$ are self-designable parameters. The deviation variable $\delta_i(lh) = x_i(lh) - \hat{x}_i(lh)$.

**Remark 1** *Considering that it takes some time to make ET judgments on agent states. When designing the DET condition* (8), *we use the agent state $q_i(\varsigma h - h)$ at the previous sampling moment, which is more logical. In addition, the agent state is taken into account when designing DET conditions, which makes the frequency of triggering not increase when the agent state is close to the optimal value. In the design of DET conditions, different parameters $\varepsilon_i$, $\mu_i$, $\omega_i$ can be set for each agent according to different needs to ensure that it can make the most accurate response in a specific situation. For the convenience of analysis, this paper chooses to set the same $\varepsilon_i$, $\mu_i$, $\omega_i$ for each agent in order to study its behavior and response ability more clearly.*

## 3.2 Main results and analysis

**Theorem 1**. *Assume that the communication graph $\mathcal{G}$ is balanced. If the following stability conditions are valid, the proposed algorithm* (6) *based on DET conditions* (8) *enables to address problem* (1), *i.e.* $\lim_{t\to\infty} x_i(t) = x^*$.

$$\frac{1}{2} - \frac{\beta_n}{2\phi_{\min}\alpha_2}\tau - \frac{\beta_n}{2\phi_{\min}\alpha_2}h - \gamma > 0, \tag{11}$$

$$h(\mu_i + \frac{\varepsilon_i}{\omega_i}) < 1, \tag{12}$$

$$\mu_i - \frac{1-\varepsilon_i}{\omega_i} > 0, \tag{13}$$

where $\phi_{\min} = \min\{\phi_1, \ldots, \phi_n\}$.

***Proof***: For $t \in [\varsigma h + \tau, (\varsigma + 1)h + \tau)$, define a Lyapunov function

$$\mathbb{V}(t) = \mathbb{V}_1(t) + \mathbb{V}_2(t). \tag{14}$$

Let $\mathbb{V}_1(t) = \sum_{i=1}^{n}(g_i(x^*) - g_i(x_i(t)) - g'_i(x_i(t))(x^* - x_i(t)))$ and $\mathbb{V}_2(t) = \sum_{i=1}^{n}\xi_i(t)$.

According to the property of convex function (2) yield

$$\mathbb{V}_1(t) \geq \sum_{i=1}^{n} \frac{\phi_i}{2}(x^* - x_i(t))^2 \geq 0. \tag{15}$$

According to (8) and (10) have

$$\dot{\xi}_i(t) \geq -\mu_i \xi_i(\varsigma h) - \frac{\varepsilon_i}{\omega_i} \xi_i(\varsigma h), \tag{16}$$

Furthermore, we obtain

$$\xi_i(\varsigma h) \geq \left(1 - h(\mu_i + \frac{\varepsilon_i}{\omega_i})\right)^{\varsigma} \xi_i(0). \tag{17}$$

Based on (12), we get $0 < 1 - h\left(\mu_i + \frac{\varepsilon_i}{\omega_i}\right) < 1$. Then, $\xi_i(\varsigma h) > 0$. Furthermore, it follows

$$\xi_i(t) \geq \xi_i(\varsigma h) - (t - \varsigma h)(\mu_i + \frac{\varepsilon_i}{\omega_i})\xi_i(\varsigma h) \geq \left(1 - h(\mu_i + \frac{\varepsilon_i}{\omega_i})\right)^{\varsigma+1} \xi_i(0). \tag{18}$$

Therefore, one obtains

$$\xi_i(t) \geq 0. \tag{19}$$

Due to (15) and (19), we can derive $\mathbb{V}(t) \geq 0$. Then we derive $\mathbb{V}_1(t)$ and $\mathbb{V}_2(t)$ yields

$$\dot{\mathbb{V}}_1(t) = \sum_{i=1}^{n} -g''_i(x_i(t))\dot{x}_i(t)(x^* - x_i(t)). \tag{20}$$

Furthermore, one has that

$$\dot{\mathbb{V}}_1(t) = \sum_{i=1}^{n} \left(\sum_{j \in N_i} a_{ij}(\hat{x}_i(t - \tau) - \hat{x}_j(t - \tau))\right) \times (x^* - x_i(t)). \tag{21}$$

Due to the following properties of undirected graphs $\mathcal{G}$.

$$\sum_{i=1}^{n} \sum_{j=1}^{n} a_{ij}(\hat{x}_j(t - \tau) - \hat{x}_i(t - \tau))x^* = 0. \tag{22}$$

Furthermore, due to $\hat{x}_i(t - \tau) = \hat{x}_i(\varsigma h)$, one has that

$$\dot{\mathcal{X}}_1 = -\mathcal{X}^T(t)\mathcal{L}\hat{\mathcal{X}}(t - \tau) = -\mathcal{X}^T(t)\mathcal{L}\hat{\mathcal{X}}(\varsigma h), \tag{23}$$

where $\mathcal{X}(t) = [x_1(t), \ldots, x_n(t)]^T, \hat{\mathcal{X}}(t) = [\hat{x}_1(t), \ldots, \hat{x}_n(t)]^T$.

Based on (6) yield

$$\begin{aligned}
\mathcal{X}(t) &= \mathcal{X}(\varsigma h + \tau) + (t - \varsigma h - \tau)\dot{\mathcal{X}}(\varsigma h + \tau) \\
&= \mathcal{X}(\varsigma h) - \tau\Gamma_2\mathcal{L}\hat{\mathcal{X}}(\varsigma h - \tau) - (t - \varsigma h - \tau)\Gamma_1\mathcal{L}\hat{\mathcal{X}}(\varsigma h) \\
&= \mathcal{X}(\varsigma h) - \tau\Gamma_2\mathcal{L}\hat{\mathcal{X}}(\varsigma h - h) - (t - \varsigma h - \tau)\Gamma_1\mathcal{L}\hat{\mathcal{X}}(\varsigma h) \\
&= \Delta(\varsigma h) + \hat{\mathcal{X}}(\varsigma h) - \tau\Gamma_2\mathcal{L}\hat{\mathcal{X}}(\varsigma h - h) - (t - \varsigma h - \tau)\Gamma_1\mathcal{L}\hat{\mathcal{X}}(\varsigma h),
\end{aligned} \tag{24}$$

where $\Gamma_1 = diag\{(g''_1(\tilde{x}_1^1))^{-1}, \ldots, (g''_1(\tilde{x}_n^1))^{-1}\}$ and $\Gamma_2 = diag\{(g''_1(\tilde{x}_1^2))^{-1}, \ldots, (g''_1(\tilde{x}_n^2))^{-1}\}$, $\tilde{x}_i^1 \in (x_i(\varsigma h + \tau), x_i(t)), \tilde{x}_i^2 \in (x_i(\varsigma h), x_i(\varsigma h + \tau)), \Delta(t) = [\delta_1(t), \delta_2(t), \ldots, \delta_n(t)]^T, i = 1, 2, \ldots, n.$

Substituting Eqs ([24]) into ([23]) yields

$$
\begin{aligned}
\dot{\mathbb{V}}_1 \leq\ & -\hat{\mathcal{X}}(\varsigma h)^T \mathcal{L} \hat{\mathcal{X}}(\varsigma h) - \Delta(\varsigma h)^T \mathcal{L} \hat{\mathcal{X}}(\varsigma h) \\
& + \tau \Gamma_2 \hat{\mathcal{X}}(\varsigma h - h)^T \mathcal{L}^T \mathcal{L} \hat{\mathcal{X}}(\varsigma h) \\
& + (t - \varsigma h - \tau) \hat{\mathcal{X}}(\varsigma h)^T \Gamma_1 \mathcal{L}^T \mathcal{L} \hat{\mathcal{X}}(\varsigma h).
\end{aligned}
\tag{25}
$$

As the inequality ([4]), there exists

$$
g_i''(x_i(t))^{-1} \leq \frac{1}{\phi_i}.
\tag{26}
$$

Furthermore, $\Gamma_1 \leq \frac{1}{\phi_{\min}} I_n$, $\Gamma_2 \leq \frac{1}{\phi_{\min}} I_n$, where $\phi_{\min} = \min\{\phi_{1,\ldots,}\phi_n\}$.
Note that

$$
\hat{\mathcal{X}}(\varsigma h)^T \mathcal{L} \hat{\mathcal{X}}(\varsigma h) = \frac{1}{2} \sum_{i=1}^{n} q_i(\varsigma h).
\tag{27}
$$

Based on Lemma 1 yield

$$
\begin{aligned}
(t - \varsigma h - \tau) \hat{\mathcal{X}}(\varsigma h)^T \mathcal{L}^T \Gamma_1 \mathcal{L} \hat{\mathcal{X}}(\varsigma h) \quad &\leq \frac{h}{\phi_{\min}} \hat{\mathcal{X}}(\varsigma h)^T \mathcal{L}^T \mathcal{L} \hat{\mathcal{X}}(\varsigma h) \\
&\leq \frac{h \beta_n}{\phi_{\min} \alpha_2} \hat{\mathcal{X}}(\varsigma h)^T \mathcal{L} \hat{\mathcal{X}}(\varsigma h) \\
&\leq \frac{h \beta_n}{\phi_{\min} \alpha_2} \frac{1}{2} \sum_{i=1}^{n} q_i(\varsigma h).
\end{aligned}
\tag{28}
$$

By utilizing Lemma 1 and Young inequality, one has

$$
\begin{aligned}
& \tau \hat{\mathcal{X}}(\varsigma h - h)^T \mathcal{L}^T \Gamma_2 \mathcal{L} \hat{\mathcal{X}}(\varsigma h) \\
&\leq \frac{\tau \beta_n}{2\phi_{\min}\alpha_2} \hat{\mathcal{X}}(\varsigma h - h)^T \mathcal{L} \hat{\mathcal{X}}(\varsigma h - h) + \frac{\tau \beta_n}{2\phi_{\min}\alpha_2} \hat{\mathcal{X}}(\varsigma h)^T \mathcal{L} \hat{\mathcal{X}}(\varsigma h) \\
&\leq \frac{\tau \beta_n}{2\phi_{\min}\alpha_2} \left[ \frac{1}{2} \sum_{i=1}^{n} q_i(\varsigma h - h) + \frac{1}{2} \sum_{i=1}^{n} q_i(\varsigma h) \right].
\end{aligned}
\tag{29}
$$

Then, we have

$$
\begin{aligned}
\dot{\mathbb{V}}_1 \leq\ & \left( \frac{h \beta_n}{2\phi_{\min}\alpha_2} + \frac{\tau \beta_n}{4\phi_{\min}\alpha_2} - \frac{1}{2} \right) \sum_{i=1}^{n} q_i(\varsigma h) \\
& + \frac{\tau \beta_n}{4\phi_{\min}\alpha_2} \sum_{i=1}^{n} q_i(\varsigma h - h) + \sum_{i=1}^{n} d_i \delta_i^2(\varsigma h).
\end{aligned}
\tag{30}
$$

Based on ([10]), we get

$$
\dot{\mathbb{V}}_2 = \sum_{i=1}^{n} \left( -\mu_i \xi_i(\varsigma h) + \varepsilon_i (\gamma q_i(\varsigma h - h) - d_i \delta_i^2(\varsigma h)) \right).
\tag{31}
$$

Then, we obtain

$$
\begin{aligned}
\dot{\mathbb{V}} \quad &\leq \dot{\mathbb{V}}_1 + \dot{\mathbb{V}}_2 \\
&\leq -\left(\frac{1}{2} - \frac{h\beta_n}{2\phi_{\min}\alpha_2} - \frac{\tau\beta_n}{4\phi_{\min}\alpha_2}\right)\sum_{i=1}^{n} q_i(\varsigma h) \\
&\quad + \frac{\tau\beta_n}{4\phi_{\min}\alpha_2}\sum_{i=1}^{n} q_i(\varsigma h - h) \\
&\quad + \sum_{i=1}^{n} d_i\delta_i^2(\varsigma h) - \sum_{i=1}^{n}\mu_i\xi_i(\varsigma h) \\
&\quad + \sum_{i=1}^{n}\varepsilon_i(\gamma q_i(\varsigma h - h) - d_i\delta_i^2(\varsigma h)).
\end{aligned}
\tag{32}
$$

Add and subtract $\sum_{i=1}^{n}\gamma q_i(\varsigma h - h)$ to right equation, one obtain

$$
\begin{aligned}
\dot{\mathbb{V}} \quad &\leq -\left(\frac{1}{2} - \frac{h\beta_n}{2\phi_{\min}\alpha_2} - \frac{\tau\beta_n}{4\phi_{\min}\alpha_2}\right)\sum_{i=1}^{n} q_i(\varsigma h) \\
&\quad + \frac{\tau\beta_n}{4\phi_{\min}\alpha_2}\sum_{i=1}^{n} q_i(\varsigma h - h) + \sum_{i=1}^{n} d_i\delta_i^2(\varsigma h) \\
&\quad - \sum_{i=1}^{n}\gamma q_i(\varsigma h - h) + \sum_{i=1}^{n}\gamma q_i(\varsigma h - h) \\
&\quad - \sum_{i=1}^{n}\mu_i\xi_i(\varsigma h) + \sum_{i=1}^{n}\varepsilon_i(\gamma q_i(\varsigma h - h) - d_i\delta_i^2(\varsigma h)) \\
&\leq -\left(\frac{1}{2} - \frac{h\beta_n}{2\phi_{\min}\alpha_2} - \frac{\tau\beta_n}{4\phi_{\min}\alpha_2}\right)\sum_{i=1}^{n} q_i(\varsigma h) \\
&\quad + \left(\frac{\tau\beta_n}{4\phi_{\min}\alpha_2} + \gamma\right)\sum_{i=1}^{n} q_i(\varsigma h - h) \\
&\quad - \left(\mu_i - \frac{1 - \varepsilon_i}{\omega_i}\right)\xi_i(\varsigma h).
\end{aligned}
\tag{33}
$$

Take the integration of $\mathbb{V}(t)$ yields

$$
\mathbb{V}((\varsigma + 1)h + \tau) - \mathbb{V}(\varsigma h + \tau) = \int_{\varsigma h + \tau}^{(\varsigma+1)h+\tau} \dot{\mathbb{V}}(t)dt,
\tag{34}
$$

for $t \in [\varsigma h + \tau, (\varsigma + 1)\,h + \tau)$.

Since the Lyapunov function is positive definite, it follows that

$$
\begin{aligned}
\mathbb{V}((\varsigma+1)h+\tau) - \mathbb{V}(\varsigma h+\tau) \leq\ & -\left(\frac{1}{2} - \frac{h\beta_n}{2\phi_{\min}\alpha_2} - \frac{\tau\beta_n}{4\phi_{\min}\alpha_2}\right)h\sum_{i=1}^{n} q_i(\varsigma h) \\
& + \left(\frac{\tau\beta_n}{4\phi_{\min}\alpha_2} + \gamma\right)h\sum_{i=1}^{n} q_i(\varsigma h - h) \\
& - \left(\mu_i - \frac{1-\varepsilon_i}{\omega_i}\right)h\xi_i(\varsigma h).
\end{aligned}
\tag{35}
$$

By accumulating $\mathbb{V}((\varsigma+1)h+\tau) - \mathbb{V}(\varsigma h+\tau)$ from $\varsigma = 1$ to $\varsigma$ yield

$$
\begin{aligned}
& \mathbb{V}((\varsigma+1)h+\tau) - \mathbb{V}(h+\tau) \\
& \leq -\left(\frac{1}{2} - \frac{h\beta_n}{2\phi_{\min}\alpha_2} - \frac{\tau\beta_n}{4\phi_{\min}\alpha_2}\right)h\sum_{i=1}^{n} q_i(\varsigma h) \\
& - \left(\frac{1}{2} - \frac{\beta_n\tau}{2\phi_{\min}\alpha_2} - \frac{\beta_n h}{2\phi_{\min}\alpha_2} - \gamma\right)h\sum_{i=1}^{n}\sum_{k=1}^{\varsigma-1} q_i(kh) \\
& + \left(\frac{\tau\beta_n}{4\phi_{\min}\alpha_2} + \gamma\right)h\sum_{i=1}^{n} q_i(0) - h\left(\mu_i - \frac{1-\varepsilon_i}{\omega_i}\right)\sum_{i=1}^{n}\sum_{k=1}^{\varsigma} \xi_i(kh).
\end{aligned}
\tag{36}
$$

while

$$
\frac{1}{2} - \frac{\beta_n}{2\phi_{\min}\alpha_2}\tau - \frac{\beta_n}{2\phi_{\min}\alpha_2}h - \gamma > 0,
\tag{37}
$$

we have

$$
\frac{1}{2} - -\frac{\beta_n}{4\phi_{\min}\alpha_2}\tau - \frac{\beta_n}{2\phi_{\min}\alpha_2}h > 0.
\tag{38}
$$

Then, we can obtain,

$$
\begin{aligned}
& \lim_{l\to\infty}(\mathbb{V}((\varsigma+1)h+\tau) - \mathbb{V}(h+\tau)) \\
& \leq -\left(\frac{1}{2} - \frac{\beta_n\tau}{2\phi_{\min}\alpha_2} - \frac{\beta_n h}{2\phi_{\min}\alpha_2} - \gamma\right)h\sum_{i=1}^{n}\sum_{\varsigma=1}^{\infty} q_i(\varsigma h) \\
& + \left(\frac{\tau\beta_n}{4\phi_{\min}\alpha_2} + \gamma\right)h\sum_{i=1}^{n} q_i(0) - h\left(\mu_i - \frac{1-\varepsilon_i}{\omega_i}\right)\sum_{i=1}^{n}\sum_{\varsigma=1}^{\infty} \xi_i(\varsigma h).
\end{aligned}
\tag{39}
$$

Given that $q_i(\varsigma h) \geq 0$ and $\mathbb{V}(t) \geq 0$, Eq (39) implies that:

$$
\lim_{\varsigma\to\infty} \sum_{i=1}^{n} q_i(\varsigma h) = 0.
\tag{40}
$$

Utilizing the definition of $q_i(\varsigma h)$ and the Laplace matrix properties, it is established that the ultimate state of each agent is equivalent to a common constant value, i.e.

$$
\lim_{t\to\infty} x_1(t) = \cdots = x_n(t) = c..
\tag{41}
$$

The second order derivative of the cost function has

$$
\begin{aligned}
\sum_{i=1}^{n}\frac{d(g'_i(x_i(t)))}{dt} &= -\sum_{i=1}^{n}\sum_{j=1}^{n}a_{ij}(\hat{x}_j(t-\tau)-\hat{x}_i(t-\tau)) \\
&= -1^{T}\mathcal{L}\hat{\mathcal{X}}(t-\tau) \\
&= 0.
\end{aligned}
\tag{42}
$$

where $1 = [1\ldots, 1]_n$.

Then, we have

$$
\sum_{i=1}^{n}g'_i(x_i(t)) = \sum_{i=1}^{n}g'_i(x_i(0)) = \sum_{i=1}^{n}g'_i(x_i^*) = G'(x^*) = 0.
\tag{43}
$$

Combining (41) and (43) yields

$$
\lim_{t\to\infty}x_1(t) = \cdots = \lim_{t\to\infty}x_n(t) = c = x^*.
\tag{44}
$$

Theorem 1 is proved.

**Corollary 1** *When the time delay is not considered and the following stability conditions hold, the proposed algorithm can effectively address problem* (1) *for any $i \in \mathcal{V}$ and $t > 0$.*

$$
\frac{1}{2} - \frac{\lambda_n}{2\phi_{\min}}h - \gamma > 0.
\tag{45}
$$

$$
h(\mu_i + \frac{\varepsilon_i}{\omega_i}) < 1,
\tag{46}
$$

$$
\mu_i - \frac{1-\varepsilon_i}{\omega_i} > 0,
\tag{47}
$$

where $\phi_{\min} = \min\{\phi_1, \ldots, \phi_n\}$, $\lambda_n = \frac{\beta_n}{\alpha_2}$. Due to the exclusion of time delay, the proof process is comparatively simpler than that of Theorem 1.

***Proof***: We adopt the identical Lyapunov function as presented in Theorem 1,

$$
\mathbb{V}(t) = \mathbb{V}_1 + \mathbb{V}_2.
\tag{48}
$$

where $\mathbb{V}_1 = \sum_{i=1}^{n}(g_i(x^*) - g_i(x_i(t)) - g'_i(x_i(t))(x^* - x_i(t)))$, $\mathbb{V}_2 = \sum_{i=1}^{n}\xi_i(t)$

According to the proof above, we can get $\dot{\mathbb{V}}_1 = -\mathcal{X}^{T}(t)\mathcal{L}\hat{\mathcal{X}}(t)$.

Based on (6) and (7) yield

$$
\begin{aligned}
\dot{\mathbb{V}}_1 &= -\left(\hat{\mathcal{X}}(t) + \Delta(t)\right)\mathcal{L}\hat{\mathcal{X}}(t) \\
&= -\hat{\mathcal{X}}^{T}(t)\mathcal{L}\hat{\mathcal{X}}(t) - [\Delta(\varsigma h) + \dot{\Delta}(\varsigma h)(t-\varsigma h)]\mathcal{L}\hat{\mathcal{X}}(t) \\
&= -\hat{\mathcal{X}}^{T}(t)\mathcal{L}\hat{\mathcal{X}}(t) - \Delta^{T}(\varsigma h)\mathcal{L}\hat{\mathcal{X}}(\varsigma h) + (t-\varsigma h)\hat{\mathcal{X}}^{T}(\varsigma h)\mathcal{L}^{T}\Gamma_3^{T}\mathcal{L}\hat{\mathcal{X}}(\varsigma h).
\end{aligned}
\tag{49}
$$

Since $t \in [\varsigma h, \varsigma h + h)$, we obtain

$$
\begin{aligned}
(t - \varsigma h)\hat{\mathcal{X}}(\varsigma h)^T \mathcal{L}^T \Gamma_3 \mathcal{L}\hat{\mathcal{X}}(\varsigma h) \quad &\leq \frac{h}{\phi_{\min}}\hat{\mathcal{X}}(\varsigma h)^T \mathcal{L}^T \mathcal{L}\hat{\mathcal{X}}(\varsigma h) \\
&\leq \frac{h\beta_n}{\phi_{\min}\alpha_2}\hat{\mathcal{X}}(\varsigma h)^T \mathcal{L}\hat{\mathcal{X}}(\varsigma h) \\
&\leq \frac{h\beta_n}{\phi_{\min}\alpha_2}\frac{1}{2}\sum_{i=1}^{n} q_i(\varsigma h).
\end{aligned}
\tag{50}
$$

where $\Gamma_3 = diag\{(g_1''(\tilde{x}_1^3))^{-1}, \ldots, (g_1''(\tilde{x}_n^3))^{-1}\}$, $\tilde{x}_i^3 \in (x_i(\varsigma h), x_i(\varsigma h + h))$, $i = 1, 2, \ldots, n$ and $\Gamma_3 \leq \frac{1}{\phi_{\min}}I_n$.

Then, we obtain

$$
\begin{aligned}
\dot{\mathbb{V}}_1 \quad &\leq -\frac{1}{2}\sum_{i=1}^{n} q_i(\varsigma h) + \sum_{i=1}^{n} d_i \delta_i^2(\varsigma h) + \frac{h\beta_n}{\phi_{\min}\alpha_2}\frac{1}{2}\sum_{i=1}^{n} q_i(\varsigma h) \\
&\leq -\frac{1}{2}\left(1 - \frac{h\beta_n}{\phi_{\min}\alpha_2}\right)\sum_{i=1}^{n} q_i(\varsigma h) + \sum_{i=1}^{n} d_i \delta_i^2(\varsigma h).
\end{aligned}
\tag{51}
$$

Then, one obtains

$$
\begin{aligned}
\dot{\mathbb{V}}_1 + \dot{\mathbb{V}}_2 \quad &\leq -\frac{1}{2}\left(1 - \frac{h\beta_n}{\phi_{\min}\alpha_2}\right)\sum_{i=1}^{n} q_i(\varsigma h) \\
&\quad + \sum_{i=1}^{n}(d_i \delta_i^2(\varsigma h) - \mu_i \xi_i(\varsigma h)) \\
&\quad + \sum_{i=1}^{n}\varepsilon_i(\gamma q_i(\varsigma h - h) - d_i \delta_i^2(\varsigma h)) \\
&\leq -\frac{1}{2}\left(1 - \frac{h\beta_n}{\phi_{\min}\alpha_2}\right)\sum_{i=1}^{n} q_i(\varsigma h) + \sum_{i=1}^{n}\gamma q_i(\varsigma h - h) \\
&\quad - \left(\mu_i - \frac{1-\varepsilon_i}{\omega_i}\right)\sum_{i=1}^{n}\xi_i(\varsigma h)
\end{aligned}
\tag{52}
$$

Based on (34), we can have

$$
\begin{aligned}
&\mathbb{V}(\varsigma h + h) - \mathbb{V}(\varsigma h) \\
&\leq -\frac{1}{2}\left(1 - \frac{h\beta_n}{\phi_{\min}\alpha_2}\right)h\sum_{i=1}^{n} q_i(\varsigma h) + h\sum_{i=1}^{n}\gamma q_i(\varsigma h - h) - \left(\mu_i - \frac{1-\varepsilon_i}{\omega_i}\right)h\sum_{i=1}^{n}\xi_i(\varsigma h).
\end{aligned}
\tag{53}
$$

Through the iterative operation of (53), we can derive

$$
\begin{aligned}
\mathbb{V}(\varsigma h + h) - \mathbb{V}(h) \leq \quad &h\gamma\sum_{i=1}^{n} q_i(0) - \frac{1}{2}\left(1 - \frac{h\beta_n}{\phi_{\min}\alpha_2}\right)h\sum_{i=1}^{n} q_i(\varsigma h) \\
&- \left(\mu_i - \frac{1-\varepsilon_i}{\omega_i}\right)h\sum_{i=1}^{n}\sum_{k=1}^{\varsigma}\xi_i(kh) \\
&- \left(\frac{1}{2} - \frac{\beta_n}{2\phi_{\min}\alpha_2}h - \gamma\right)h\sum_{i=1}^{n}\sum_{k=1}^{\varsigma-1} q_i(kh).
\end{aligned}
\tag{54}
$$

Then we use the iterative method that is similar to Theorem 1 and finally get the result of Corollary 1.

**Remark 2** $\lambda_n$ *is a global information required by each agent. The following inequality gives the bound on* $\lambda_n$*. Then, the bounds for h and $\gamma$ can be obtained from* Eq (45).

$$\lambda_n \leq 2d_{\max} \leq 2(n-1), \tag{55}$$

*Thus, condition* (45) *can be changed as*:

$$\frac{1}{2} - \frac{n-1}{\phi_{\min}}h - \gamma > 0, \tag{56}$$

## 3.3 The distributed DET-ZGS algorithm with arbitrary sampling period

According to the above analysis, the sampling period significantly affects the performance of the algorithm. When the period interval is selected to be large, it cannot be guaranteed that the algorithm will find the global optimal value. This subsection handles the sampling period on the basis of algorithm (6) and proposes a distributed DET-ZGS algorithm where the sampling period can be designed arbitrarily.

The algorithm is designed as:

$$\dot{x}_i(t) = -\frac{1}{h}(\nabla^2 g_i(x_i(t)))^{-1}u_i(t), \tag{57}$$

$$u_i(t) = \sum_{j=1}^{n} a_{ij}(\hat{x}_j(t-\tau) - \hat{x}_i(t-\tau)). \tag{58}$$

The DET condition changes as

$$\omega_i(d_i\delta_i^2(\zeta h) - \gamma q_i(\zeta h - h)) \leq \xi_i(\zeta h), \tag{59}$$

$$q_i(t) = \sum_{j=1}^{n} a_{ij}(\hat{x}_j(t-\tau) - \hat{x}_i(t-\tau))^2, \tag{60}$$

$$\dot{\xi}_i(t) = \frac{1}{h}\left[-\mu_i\xi_i(\zeta h) + \varepsilon_i\big(\gamma q_i(\zeta h - h) - d_i\delta_i^2(\zeta h)\big)\right]. \tag{61}$$

**Theorem 2** *Suppose the undirected graph is connected. For $\forall i \in \mathcal{V}$ and $t > 0$, when condition* (62)–(64) *is satisfied, the system state converges to the optimal solution $x^*$ under the effect of algorithm* (57) *and the DET condition* (59).

$$\frac{1}{2} - \frac{\beta_n}{2\phi_{min}\alpha_2} - \frac{\tau\beta_n}{2\phi_{min}\alpha_2 h} - \gamma > 0, \tag{62}$$

$$\mu_i + \frac{\varepsilon_i}{\omega_i} < 1, \tag{63}$$

$$\mu_i - \frac{1-\varepsilon_i}{\omega_i} > 0. \tag{64}$$

***Proof***: For $t \in [\varsigma h + \tau, (\varsigma + 1) h + \tau)$, define a Lyapunov function

$$\mathbb{V}(t) = \mathbb{V}_1(t) + \mathbb{V}_2(t). \tag{65}$$

where $\mathbb{V}_1(t) = \sum_{i=1}^{n} (g_i(x^*) - g_i(x_i(t)) - g'_i(x_i(t))(x^* - x_i(t)))$ and $\mathbb{V}_2(t) = \sum_{i=1}^{n} \xi_i(t)$.

Then we have

$$
\begin{aligned}
\dot{\mathbb{V}}_1(t) \quad &= \frac{1}{h} \sum_{i=1}^{n} u_i(t)(x^* - x_i(t)) \\
&= -\frac{1}{h} \mathcal{X}^T(t) \mathcal{L} \hat{\mathcal{X}}(\varsigma h) \\
&= -\frac{1}{h} \Delta^T(\varsigma h) \mathcal{L} \hat{\mathcal{X}}(\varsigma h) - \frac{1}{h} \hat{\mathcal{X}}^T(\varsigma h) \mathcal{L} \hat{\mathcal{X}}(\varsigma h) \\
&\quad + \frac{\tau}{h^2} \Gamma_2 \hat{\mathcal{X}}^T(\varsigma h - h) \mathcal{L}^T \mathcal{L} \hat{\mathcal{X}}(\varsigma h) \\
&\quad + \frac{(t - \varsigma h - \tau)}{h^2} \Gamma_1 \hat{\mathcal{X}}^T(\varsigma h) \mathcal{L}^T \mathcal{L} \hat{\mathcal{X}}(\varsigma h).
\end{aligned}
\tag{66}
$$

According to (28) and (29), we obtain

$$
\begin{aligned}
\dot{\mathbb{V}}_1(t) \quad &\le \left( \frac{\beta_n}{2\phi_{min}\alpha_2} + \frac{\tau\beta_n}{4\phi_{min}\alpha_2 h} - \frac{1}{2} \right) \frac{1}{h} \sum_{i=1}^{n} q_i(\varsigma h) \\
&\quad + \frac{\tau\beta_n}{4\phi_{min}\alpha_2 h} \sum_{i=1}^{n} q_i(\varsigma h - h) + \frac{1}{h} \sum_{i=1}^{n} d_i \delta_i^2(\varsigma h).
\end{aligned}
\tag{67}
$$

From (61), it follows that

$$
\dot{\mathbb{V}}_2(t) = -\sum_{i}^{n} \left[ -\frac{\mu_i}{h} \xi_i(\varsigma h) + \frac{\varepsilon_i}{h} \left( \gamma q_i(\varsigma h - h) - d_i \delta_i^2(\varsigma h) \right) \right]
\tag{68}
$$

Combine (67) and (68) yields

$$
\begin{aligned}
\mathbb{V}(t) \le \quad &- \left( \frac{1}{2} - \frac{\beta_n}{2\phi_{min}\alpha_2} - \frac{\tau\beta_n}{4\phi_{min}\alpha_2 h} \right) \frac{1}{h} \sum_{i=1}^{n} q_i(\varsigma h) \\
&+ \frac{\tau\beta_n}{4\phi_{min}\alpha_2 h^2} \sum_{i=1}^{n} q_i(\varsigma h - h) \\
&+ \frac{\gamma}{h} \sum_{i=1}^{n} q_i(\varsigma h - h) \\
&- \frac{1}{h} \sum_{i=1}^{n} (\mu_i - \frac{1 - \varepsilon_i}{\omega_i}) \xi_i(\varsigma h).
\end{aligned}
\tag{69}
$$

From Eq (34), it follows that

$$
\begin{aligned}
\mathbb{V}((\varsigma+1)h+\tau) - \mathbb{V}(\varsigma h+\tau) \leq \quad & -\left(\frac{1}{2} - \frac{\beta_n}{2\phi_{min}\alpha_2} - \frac{\tau\beta_n}{4\phi_{min}\alpha_2 h}\right)\sum_{i=1}^{n} q_i(\varsigma h) \\
& + \frac{\tau\beta_n}{4\phi_{min}\alpha_2 h}\sum_{i=1}^{n} q_i(\varsigma h - h) + \gamma\sum_{i=1}^{n} q_i(\varsigma h - h) \\
& - \sum_{i=1}^{n}(\mu_i - \frac{1-\varepsilon_i}{\omega_i})\xi_i(\varsigma h).
\end{aligned} \tag{70}
$$

Then iterate to obtain

$$
\begin{aligned}
\mathbb{V}((\varsigma+1)h+\tau) - \mathbb{V}(h+\tau) \leq \quad & -\left(\frac{1}{2} - \frac{\beta_n}{2\phi_{min}\alpha_2} - \frac{\tau\beta_n}{2\phi_{min}\alpha_2 h} - \gamma\right)\sum_{k=1}^{\varsigma-1}\sum_{i=1}^{n} q_i(kh) \\
& + \frac{\tau\beta_n}{4\phi_{min}\alpha_2 h}\sum_{i=1}^{n} q_i(0) \\
& - \left(\frac{1}{2} - \frac{\beta_n}{2\phi_{min}\alpha_2} - \frac{\tau\beta_n}{4\phi_{min}\alpha_2 h}\right)\sum_{i=1}^{n} q_i(\varsigma h) \\
& - \sum_{k=1}^{\varsigma}\sum_{i=1}^{n}\left(\mu_i - \frac{1-\varepsilon_i}{\omega_i}\right)\xi_i(kh).
\end{aligned} \tag{71}
$$

Then use the previous similar analysis to complete the proof of Theorem 2.

**Remark 3** *The stability conditions derived from Theorem 2 reveal that when the sampling period h is chosen to be small, $\tau$ and h can be eliminated due to $0 < \tau < h$; When the sampling period h takes a large value, the term $\frac{\tau\beta_n}{2\phi_{\min}\alpha_2 h}$ can be ignored. From the perspective of simulation experiments, longer sampling periods require more time steps, but still lead to optimal solutions. However, in [26], excessively large sampling periods make it difficult to satisfy the stability conditions, resulting in the algorithm's failure to converge and find the optimal solution. Therefore, the algorithm proposed in this paper allows the sampling period to be arbitrarily large, which is of great significance.*

## 4 Simulation result

In this section, a MAS comprising 4 agents is examined. The communication topology among these agents is both strongly connected and undirected, as depicted in Fig 1. Notably, each agent features a unique cost function, which is defined as follows:

$$
g_i(x) = (x-i)^4 + 4i(x-i)^2, i = 1, 2, 3, 4.
$$

It is clear that the minimum local objective function is obtained when $x_i$ is equal to $i$, respectively. However, this is not the global optimal solution $x^*$. Thus we need to employ the proposed algorithm to find $x^*$. We set the important parameters for the simulation at first. Set $\phi_{min} = 16$, $\gamma = 0.1$ and $\omega_i = 1$, $\xi_i(0) = 5$, $\mu_i = 0.2$, $\varepsilon_i = 0.2$, $x_i(0) = i$, $i = 1, 2, 3, 4$.

First, the effect of the sampling period is not considered, i.e., set $h = 1s$ and $\tau = 0.2s$, the process of agents gradually reaching consensus is shown in Fig 2. Each agent gradually agrees from the local optimal solution and finds the global optimal solution $x^* = 2.86$. This proves that the algorithm in this paper has good performance and that it is feasible to construct the DET condition using the previous moment state values. The state evolution of the control

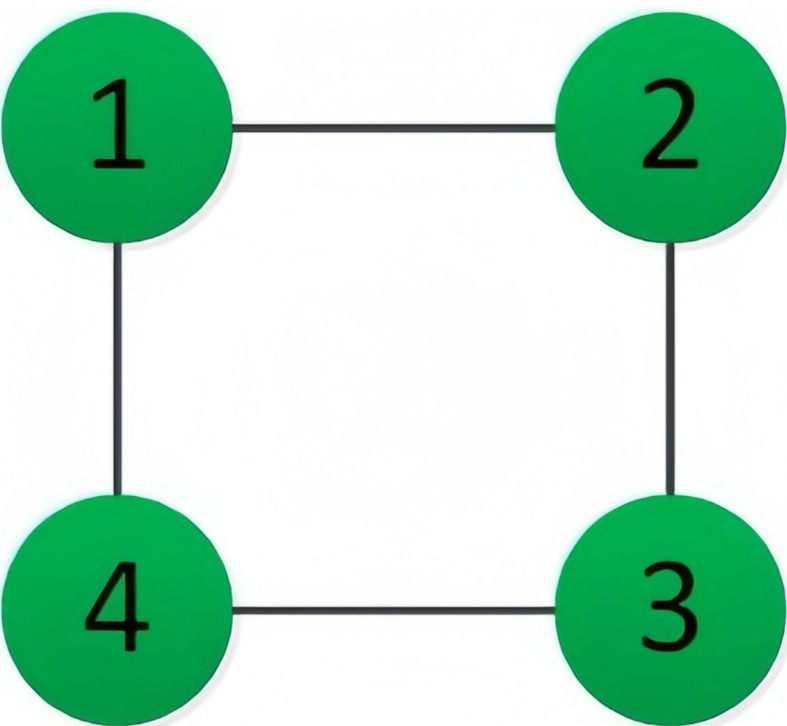

**Fig 1. The communication topology of 4 agents, where there lacks communication between agent 2 and 4, agent 1 and 3, and the rest are undirected communication.**

input $u_i$ is shown in Fig 3. It can see that $u_i$ is updated only at the interval of the ET of the agent or its neighbors and converges to 0 after reaching consensus.

When the effect of sampling period is considered, the agents' state are exhibited in Fig 4(a). A comparison with Fig 2 reveals that the agents require less time to attain the optimal value. It

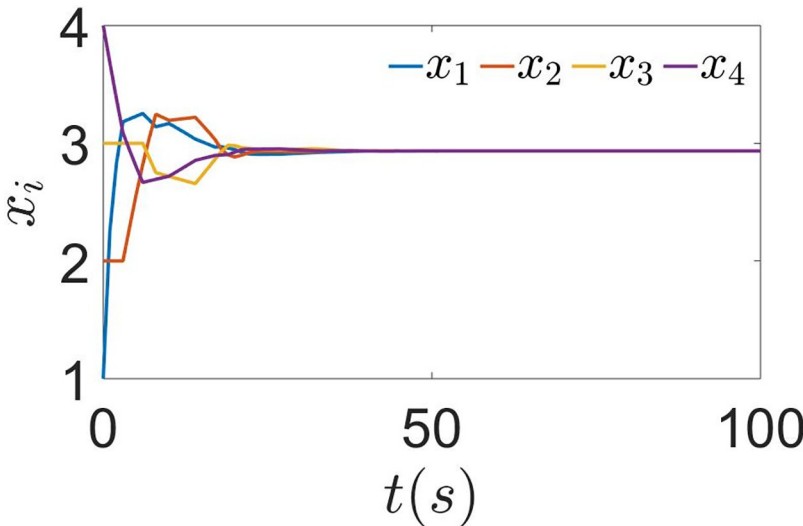

**Fig 2. The state of agent using (6) with $h = 1s$, $\tau = 0.2s$.**

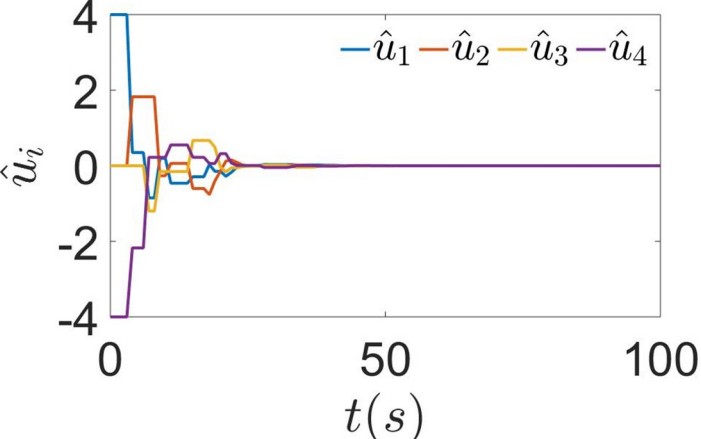

**Fig 3. The evolution of $u_i(t)$ with $h = 1s$, $\tau = 0.2s$.** $u_i(t)$ is updated when the agent or its neighbors reach the ET condition and converges to 0 after consensus is reached between agents.

is evident that the algorithm (57) outperforms algorithm (6) after the effect of sampling period is removed. The ET interval for each agent is shown in Fig 5(a). Compared to continuous communication, using an ET scheme can significantly reduce the frequency of communication. To further analyze the effect of sampling period, different sampling periods are chosen with constant time delay. When setting $h = 2s$, $\tau = 0.2s$, the agents' state are shown in Fig 4(b). Compared with Fig 4(a), it takes longer to reach the optimal solution. Then the ET interval for each agent is shown in Fig 5(b). The ET frequency is reduced after selecting a larger $h$. In fact, this is quite normal. Because a larger $h$ will weaken the feedback gain and reduce the number of control updates in a fixed finite time interval. Thus, a balance must be made between the speed of convergence and the frequency of communication.

## 5 Conclusions

This paper proposes a distributed DET-ZGS algorithm to solve the optimization problem under undirected graphs. The DET condition includes a variable related to the agent state at the previous moment, which allows to avoid continuous communication when the agent state

(a) (b)

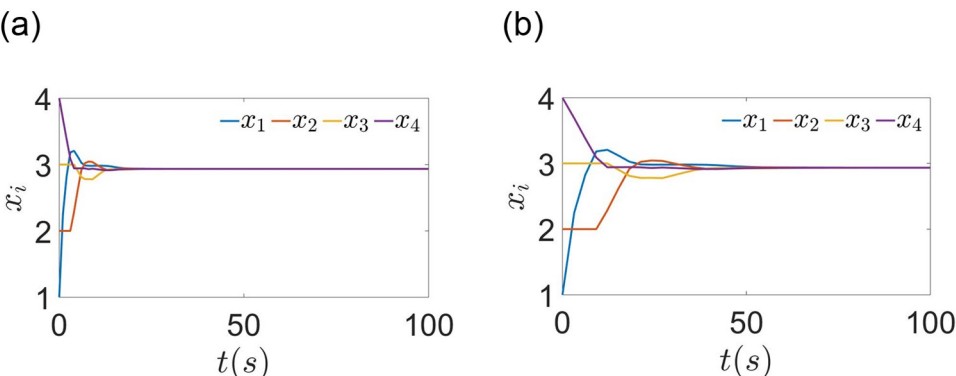

**Fig 4.** (a)The state of agent using (57) with $h = 1s$, $\tau = 0.2s$. (b) The state of agent using (57) with $h = 2s$, $\tau = 0.2s$.

(a)

(b)

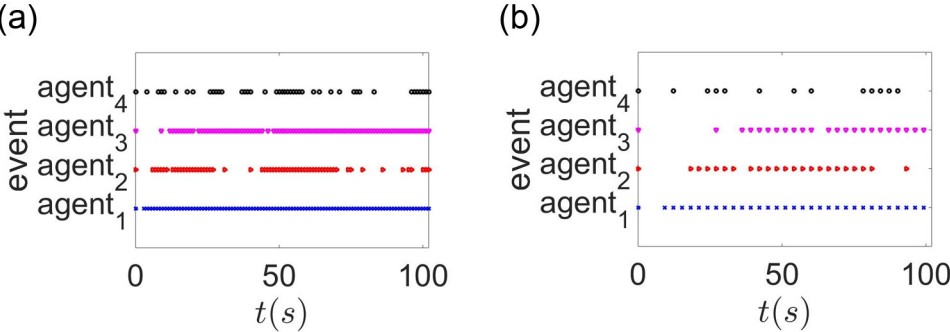

**Fig 5.** (a)The event triggered moment with $h = 1s$, $\tau = 0.2s$. (b) The event triggered moment with $h = 2s$, $\tau = 0.2s$.

is close to the optimal value. By integrating sampling method and DET mechanism, the zeno behavior can be eliminated. Furthermore, the proposed algorithm design accounts for the impact of the sampling period, which can be arbitrarily large. For systems with time delay, the sufficient conditions are presented to ensure that the MAS obtains an optimal solution. Future work will expand the algorithm to directed and switched topologies.

## Author Contributions

**Conceptualization:** Lu Jiang, Zhongyuan Zhao.

**Formal analysis:** Lunchao Xia.

**Funding acquisition:** Zhongyuan Zhao.

**Investigation:** Lu Jiang, Lunchao Xia.

**Methodology:** Lu Jiang, Lunchao Xia, Zhongyuan Zhao.

**Writing – original draft:** Lu Jiang, Lunchao Xia, Zhongyuan Zhao.

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
