## [Decision Letter · Decision Letter 0]

2 Feb 2024

PONE-D-24-01551Distributed dynamic event-triggered algorithm for optimization problem with time delayPLOS ONE

Dear Dr. Zhao,

Thank you for submitting your manuscript to PLOS ONE. After careful consideration, we feel that it has merit but does not fully meet PLOS ONE’s publication criteria as it currently stands. Therefore, we invite you to submit a revised version of the manuscript that addresses the points raised during the review process. Please submit your revised manuscript by Mar 18 2024 11:59PM. If you will need more time than this to complete your revisions, please reply to this message or contact the journal office at plosone@plos.org. Please include the following items when submitting your revised manuscript:A rebuttal letter that responds to each point raised by the academic editor and reviewer(s). You should upload this letter as a separate file labeled 'Response to Reviewers'.A marked-up copy of your manuscript that highlights changes made to the original version. You should upload this as a separate file labeled 'Revised Manuscript with Track Changes'.An unmarked version of your revised paper without tracked changes. You should upload this as a separate file labeled 'Manuscript'.If applicable, we recommend that you deposit your laboratory protocols in protocols.io to enhance the reproducibility of your results. Protocols.io assigns your protocol its own identifier (DOI) so that it can be cited independently in the future. For instructions see: https://journals.plos.org/plosone/s/submission-guidelines#loc-laboratory-protocols. Additionally, PLOS ONE offers an option for publishing peer-reviewed Lab Protocol articles, which describe protocols hosted on protocols.io. Read more information on sharing protocols at https://plos.org/protocols?utm_medium=editorial-email&utm_source=authorletters&utm_campaign=protocols.

We look forward to receiving your revised manuscript.

Kind regards,

Gang Wang

Academic Editor

PLOS ONE

Journal Requirements:

“"This research was supported by the National Natural Science Foundation of China (No. 246

U23B2061, 62033010), the Natural Science Foundation of Jiangsu Province, China (No. 247

BK20200824) and Postgraduate Research and Practice Innovation Program of Jiangsu 248

Province, China (No. SJCX23 0391)."       

“This research was supported by the National Natural Science Foundation of China (No. 246

U23B2061, 62033010), the Natural Science Foundation of Jiangsu Province, China (No. 247

BK20200824) and Postgraduate Research and Practice Innovation Program of Jiangsu 248

Province, China (No. SJCX23 0391).”

"This research was supported by the National Natural Science Foundation of China (No. 246

U23B2061, 62033010), the Natural Science Foundation of Jiangsu Province, China (No. 247

BK20200824) and Postgraduate Research and Practice Innovation Program of Jiangsu 248

Province, China (No. SJCX23 0391)."       

Additional Editor Comments :

Two reviews are obtained for this submission. Both reviewers are positive about this work, but one of them suggests that it can be improved by providing more explanation on the implementation process of the proposed optimization algorithm and the meanings of the variables. The authors are encouraged to address these comments appropriately.

Reviewers' comments:

Reviewer's Responses to Questions

**Comments to the Author**

1. Is the manuscript technically sound, and do the data support the conclusions?

Reviewer #1: Yes

Reviewer #2: Yes

2. Has the statistical analysis been performed appropriately and rigorously? 

Reviewer #1: Yes

Reviewer #2: Yes

3. Have the authors made all data underlying the findings in their manuscript fully available?

Reviewer #1: Yes

Reviewer #2: Yes

4. Is the manuscript presented in an intelligible fashion and written in standard English?

Reviewer #1: Yes

Reviewer #2: Yes

5. Review Comments to the Author

Reviewer #1: The manuscript provides a comprehensive study on the optimization problem of MAS under undirected graphs, which is a relevant and important research area. The use of a zero-gradient-sum (ZGS) algorithm with a dynamic event-triggered (DET) mechanism is novel and promising. It effectively reduces the reliance on continuous state information from neighbors, which can lead to improved efficiency in multi-agent systems. The following comments should be taken into account for further improvement.

1. The paper proposes a method to optimize multi-agent systems under undirected graphs by reducing the communication frequency among agents. However, the description of the implementation process of the proposed optimization algorithm is not sufficiently thorough. It is recommended to provide a more detailed and specific description.

2. The meanings of the variables used in the formulas are not sufficiently explained, such as the specific reference of positive parameters related to agent . It is recommended that detailed explanations and definitions be provided in the manuscript.

3. It is better to give a specific definition and explanation of "allowing the sampling period to be arbitrarily large".

Reviewer #2: This paper studied the optimization problem of multi-agent systems. The zero-gradient-sum (ZGS) algorithm based on dynamic event-triggered (DET) mechanism is investigated to reduce the communication frequency among agents. This paper can be accepted.

6. PLOS authors have the option to publish the peer review history of their article (what does this mean?). If published, this will include your full peer review and any attached files.

Reviewer #1: No

Reviewer #2: No

---

## [Editor Report · Decision Letter 1]

13 Feb 2024

Distributed dynamic event-triggered algorithm for optimization problem with time delay

PONE-D-24-01551R1

Dear Dr. Zhao,

We’re pleased to inform you that your manuscript has been judged scientifically suitable for publication and will be formally accepted for publication once it meets all outstanding technical requirements.

Kind regards,

Gang Wang

Academic Editor

PLOS ONE

Additional Editor Comments (optional):

This revised paper can be accepted.
---

## [Editor Report · Acceptance letter]

8 Apr 2024

PONE-D-24-01551R1 

PLOS ONE

Dear Dr. Zhao, 

I'm pleased to inform you that your manuscript has been deemed suitable for publication in PLOS ONE. Congratulations! Your manuscript is now being handed over to our production team.

Kind regards, 

on behalf of

Dr. Gang Wang 

Academic Editor

PLOS ONE